# Association of Coffee and Tea Intake with Bone Mineral Density and Hip Fracture: A Meta-Analysis

**DOI:** 10.3390/medicina59061177

**Published:** 2023-06-20

**Authors:** Chun-Ching Chen, Yu-Ming Shen, Siou-Bi Li, Shu-Wei Huang, Yi-Jie Kuo, Yu-Pin Chen

**Affiliations:** 1Department of General Medicine, Changhua Christian Hospital, Changhua 500, Taiwan; 2Department of General Medicine, Shuang Ho Hospital, Taipei Medical University, Taipei 234, Taiwan; 3Department of Orthopedics, Wan Fang Hospital, Taipei Medical University, Taipei 116, Taiwan; 4Department of Orthopedics, School of Medicine, College of Medicine, Taipei Medical University, Taipei 116, Taiwan

**Keywords:** coffee, tea, bone mineral density, osteoporotic fractures, osteoporosis

## Abstract

*Background and Objectives*: Osteoporosis is characterized by low bone mass and high bone fragility. Findings regarding the association of coffee and tea intake with osteoporosis have been inconsistent. We conducted this meta-analysis to investigate whether coffee and tea intake is associated with low bone mineral density (BMD) and high hip fracture risk. *Materials and Methods*: PubMed, MEDLINE, and Embase were searched for relevant studies published before 2022. Studies on the effects of coffee/tea intake on hip fracture/BMD were included in our meta-analysis, whereas those focusing on specific disease groups and those with no relevant coffee/tea intake data were excluded. We assessed mean difference (MD; for BMD) and pooled hazard ratio (HR; for hip fracture) values with 95% confidence interval (CI) values. The cohort was divided into high- and low-intake groups considering the thresholds of 1 and 2 cups/day for tea and coffee, respectively. *Results*: Our meta-analysis included 20 studies comprising 508,312 individuals. The pooled MD was 0.020 for coffee (95% CI, −0.003 to 0.044) and 0.039 for tea (95% CI, −0.012 to 0.09), whereas the pooled HR was 1.008 for coffee (95% CI, 0.760 to 1.337) and 0.93 for tea (95% CI, 0.84 to 1.03). *Conclusions*: Our meta-analysis results suggest that daily coffee or tea consumption is not associated with BMD or hip fracture risk.

## 1. Introduction

With the increasing age of the global population, the prevalence of osteoporosis is expected to substantially increase in the next few decades [1]. Osteoporosis is a major public health concern with increasing social and economic burden worldwide. This disease is associated with severe complications, such as osteoporotic fracture and fracture-related functional disability [1]. Therefore, risk factors for osteoporosis must be identified and validated at the earliest. Factors affecting the risk of osteoporosis include inadequate physical activity, low body mass index [2], inadequate intake of calcium [3], vitamin D deficiency [4], smoking [5], alcohol intake [5], and dietary coffee and tea intake [6]. Coffee and tea are the main dietary sources of caffeine, which is a psychoactive substance used widely across the globe [7]. Caffeine-containing products may affect the cardiovascular and central nervous systems [8] through various biological pathways. Several studies have indicated that a high dietary intake of caffeinated beverages is associated with decreased bone mineral density (BMD) [9,10]. Some cohort studies have reported that the intake of coffee and tea is associated with a considerable risk of hip fracture [11,12,13,14,15,16,17,18,19,20,21,22,23,24,25,26]. However, several studies indicated different opinions. A meta-analysis conducted by Sheng et al., indicated that although coffee intake is not associated with hip fracture risk, tea intake is associated with a reduction in hip fracture risk [6]. In another meta-analysis, Zeng et al., demonstrated that a higher level of daily coffee intake is strongly associated with a lower risk of osteoporosis [27]. However, they did not precisely mention the threshold used to define high- and low-coffee intake groups or clarify the association of coffee and tea intake with BMD; the possible association is yet to be validated. Additional studies must be conducted to investigate whether the intake of these two types of caffeinated beverages increases hip fracture risk and reduces BMD. Based on meta-analysis of prospective cohort and cross-sectional studies, we investigated the association of coffee and tea intake with BMD and hip fracture risk.

## 2. Materials and Methods

### 2.1. Literature Search

Without imposing any restrictions, we searched the PubMed, Embase, and MEDLINE databases for relevant studies published from the databases’ inception dates to July 2022. Medical Subject Headings (MeSH) search strategies (Appendix A) were adopted; search terms related to outcome variables (e.g., BMD, osteoporosis, and fracture) and influencing factors (e.g., caffeine, coffee, or tea) were combined. After identifying relevant studies, we reviewed their full texts and reference lists. This study was registered with the International Prospective Register of Systematic Reviews (CRD356643).

### 2.2. Study Selection

Studies were included in the meta-analysis if they were (1) observational studies (cohort or case-control), (2) focused on coffee or tea intake, (3) included outcomes that are of interest for our study (i.e., BMD measured through dual-energy X-ray absorptiometry [DXA] and hip fracture risk), (4) offered information on the frequency and amount of coffee or tea intake, and (5) reported accurate data pertaining to BMD or hazard ratios (HRs) for hip fractures (including 95% confidence intervals [CIs]). Studies not fulfilling the aforementioned inclusion criteria (e.g., studies on mixed beverages and those not differentiating between the effects coffee and tea intake) were excluded from our analysis. If multiple articles were discovered to have originated from a single study, only the article offering the highest volume of information was included.

### 2.3. Data Extraction

Two reviewers independently evaluated retrieved studies by using a standardized data collection form (Appendix A). Discrepancies in judgment were resolved through discussion and by referring to original studies. A third reviewer assisted in resolving disagreements regarding the abstracted data. From the studies included in the final analysis, we collected the following data: first author’s surname, publication year, country, study design, study period, follow-up duration (for cohort studies), population size (number of individuals in the case group, control group, and total cohort), participants’ age and sex, exclusion criteria for participants, measurements, exposure range, outcome variables (BMD or hip fracture), and covariates.

### 2.4. Study Quality Assessment

Study quality was assessed using the Newcastle–Ottawa Scale [28] for cohort studies and the Joanna Briggs Institute Critical Appraisal Checklist [29] for cross-sectional studies. Scores on the Newcastle–Ottawa Scale ranged from 0 to 9; although this scale does not impose a cutoff for differentiating between high- and low-quality studies, a variable score of ≥7 has previously been used as the cutoff for identifying high-quality studies [30]. Accordingly, in the present study, studies with scores of ≥7 and <7 were categorized as high- and low-quality studies, respectively.

### 2.5. Statistical Analyses

#### 2.5.1. BMD Group

We used BMD data to investigate the association of coffee and tea intake with the risk of osteoporosis and obtained relevant mean difference (MD) values. To account for within- and between-study variations [31], we used low-level coffee or tea intake as the reference and high-level coffee or tea intake as the exposure event. The thresholds used for differentiating between high- and low-intake groups were 1 and 2 cups/day for tea and coffee, respectively.

#### 2.5.2. Hip Fracture Group

We obtained HRs to investigate the association of coffee and tea intake with the risk of hip fracture. From the included studies, we collected data regarding high- and low-level coffee and tea intake on the basis of the aforementioned thresholds.

Random-effects models were used for meta-analysis. Heterogeneity among the included studies was assessed using Cochran’s Q test at a significance level of *p*  <  0.10, and I2 statistics [32]. I2 was used to measure potential inconsistency, which was calculated as follows:I2 = 100% × (Q − df)/Q

Here, df is the degree of freedom and an I2 value of ≥50% indicates significant heterogeneity. Subsequently, subgroup analyses were performed to determine the effects of factors causing heterogeneity (e.g., age at screening, particularly for postmenopausal women, sex, geographical location, adjustment for calcium intake, and adjustment for tea or coffee intake). Sensitivity analysis was performed for both BMD- and hip fracture-related studies to identify the potential sources of heterogeneity. In addition, publication bias was investigated through funnel plot asymmetry tests and Egger’s and Begg’s regression tests [33]. All analyses were performed using Comprehensive Meta-Analysis (version 3.0; Biostat, NJ, USA). Statistical significance was set at *p* < 0.05, unless mentioned otherwise.

## 3. Results

### 3.1. Study Selection

A total of 413 studies were retrieved from Pubmed (n = 131), MEDLINE (n = 19), and Embase (n = 263), and 73 studies were excluded for duplication. After the titles and abstracts of the remaining studies were reviewed, 220 records were excluded. Subsequently, eight studies were discarded because of the lack of full-text content. Finally, 106 studies were examined in the final assessment.

In the final assessment, 86 studies were excluded for the following reasons: inadequate data (20 studies), BMD measurements were not obtained through DXA (10 studies), mismatched criteria (six studies), dietary caffeine consumption data did not distinguish between caffeine from coffee and tea (nine studies), format of outcome was inconsistent with the requirements of the present study (36 studies), inconsistent application of cutoff point within a study (four studies), and only a specific population under dialysis was examined (one study). Finally, 20 studies, of which 10 reported BMD data and the other 10 reported hip fracture HR data, were included for further analysis (Figure 1).

### 3.2. Study Characteristics

Table 1 summarizes the studies included in our meta-analysis. In total, 416,847 individuals (BMD-related studies, 99,750 individuals; hip fracture–related studies, 408,562 individuals) were included in the final analysis. The included studies were published between 1991 and 2021 and had a sample size of 138–4979 individuals. Of the studies, 10 were on coffee intake [14,18,20,21,22,25,26,34,35,36], whereas 11 were on tea intake [11,24,25,26,37,38,39,40,41,42,43]. Regarding population, 10 studies included only postmenopausal women, whereas four studies included both women and men. BMD-related studies were conducted in the following countries or regions: United States (n = 2), Europe (n = 2; one in Sweden and one in the United Kingdom), China (n = 5), and Australia (n = 1). BMD analysis was adjusted in nine studies, but the covariates varied across the studies. Hip fracture–related studies were conducted in the following countries or regions: United States (n = 2), Europe (n = 6; three in Sweden, one in Finland, one in Norway, and one in the Netherlands), Asia (n = 2; one in Singapore and one in China), and Australia (n = 1).

### 3.3. Quality Assessment Results

Among the cohort studies, two had six points, four had seven points, four had eight points, and one had nine points; thus, all included cohort studies were of higher-than-moderate (i.e., acceptable) quality (Appendix A).

Among the cross-sectional studies, one had five points, four had seven points, and two had eight points; thus, all included cross-sectional studies were of higher-than-moderate quality (Appendix A).

### 3.4. Meta-Analysis Results

#### 3.4.1. Results Pertaining to BMD

The meta-analysis results of BMD-related studies stratified by beverages are presented below. Figure 2 and Figure 3 present the statistics of studies regarding the effects of coffee and tea, respectively, on BMD.

Coffee

Two studies regarding the effects of coffee on BMD included only postmenopausal women, whereas one study included both men and women. We divided the total number of participants into high- and low-intake groups using 2 cups/day as the threshold. Regarding heterogeneity among the study populations, an analysis of the data obtained from studies including postmenopausal women revealed a nonsignificant association between coffee intake and BMD (MD = 0.020, 95% CI = −0.003 to 0.044, and *p* = 0.093; I2 = 0% and *p* = 0.937).

Tea

All seven studies regarding the effects of tea on BMD included only postmenopausal women [11,37,38,39,40,41,42]. The association of tea intake with BMD was nonsignificant in this population (MD = 0.039, 95% CI = 0.012–0.09, and *p* = 0.132). Heterogeneity among the studies was high (I2 = 98.455% and *p* < 0.001).

#### 3.4.2. Results Pertaining to Hip Fracture

The meta-analysis results of hip fracture-related studies stratified by beverage are presented below. Figure 4 and Figure 5 present the statistics of studies regarding the effects of coffee and tea, respectively, on hip fracture risk.

Coffee

Seven studies investigated the effects of coffee on hip fracture risk [14,17,20,21,23,25,26]. The pooled HR for individuals with high levels of coffee intake was 1.008 (95% CI = 0.760–1.337 and *p* = 0.957; I2 = 83.95% and *p* < 0.001). No strong association was noted between coffee intake and hip fracture risk.

Tea

Four studies evaluated the effects of tea on hip fracture risk [11,24,25,26]. The pooled HR for individuals with high levels of tea intake was 0.929 (95% CI = 0.836–1.033 and *p* = 0.172; I2 = 0% and *p* = 0.463). No strong association was noted between tea intake and hip fracture risk.

### 3.5. Sensitivity Analysis

A sensitivity analysis was performed to investigate whether the results of any one study exerted disproportionate effects on the overall results. The results of the included studies were stable and did not exhibit considerable changes; thus, the rationality and reliability of our analysis were validated.

### 3.6. Publication Bias

#### 3.6.1. Results Pertaining to BMD

The funnel plots depicted in Appendix A indicate that most studies were within the 95% CI values and exhibited prominent symmetry. The results of Egger’s and Begg’s tests revealed no publication biases in studies on coffee intake (Egger’s test: t = 3.018 and *p* = 0.204; Begg’s test: Z = 1.567 and *p* = 0.117) or tea intake (Egger’s test: t = 0.892 and *p* = 0.413; Begg’s test: Z = 0.9 and *p* = 0.367).

#### 3.6.2. Results Pertaining to Hip Fracture

Relevant funnel plots are depicted in Appendix A. The results of Egger’s and Begg’s tests revealed no publication biases in studies on coffee intake (Egger’s vs. Begg’s test, *p* = 0.730 vs. 0.55, respectively) or tea intake (Egger’s vs. Begg’s test, *p* = 0.272 vs. 0.734, respectively).

## 4. Discussion

Overall, our findings revealed no strong association between tea or coffee intake and BMD or hip fracture.

### 4.1. Effects of Coffee on Bone Health

Although the mechanism through which coffee increases the risk of osteoporosis is unclear, Xu et al., indicated that the caffeine in coffee is a major factor that contributes to a reduction in BMD [44]. Several prospective cohort studies have suggested that an increased intake of caffeinated beverages is associated with an increased risk of osteoporosis and hip fracture, particularly in postmenopausal women. An in vitro animal study reported that caffeine adversely influences BMD by affecting adenosine receptors, thereby inhibiting bone formation and accelerating bone resorption [45]. In another animal study, caffeine promoted osteoclast differentiation and reduced BMD in growing rats [46]. In a human study [47], caffeine affected bone metabolism by increasing the urinary excretion of calcium, reducing the expression of vitamin D receptors, and reducing the 1,25(OH)2D3-induced alkaline phosphatase activity of osteoblasts.

Some studies have reported findings that are inconsistent with those of the aforementioned studies. In their review study, Dew et al. [48] highlighted that in several studies, the association between caffeine intake and low BMD became nonsignificant after covariate adjustment. Nawrot et al., demonstrated that the BMD of individuals was not markedly affected by a daily caffeine intake of <400 mg if their daily calcium intake was a minimum of 800 mg [49]. Thus, adequate calcium intake may increase the threshold at which caffeine intake affects bone health. Massey et al. [50] indicated that compared with young adults, older adults are less capable of adapting calcium absorption to counteract the caffeine-induced urinary loss of calcium. Coffee has multiple constituents. Although caffeine is the most frequently studied constituent of coffee, others such as tannic acid should not be neglected. An animal study revealed that tannic acid in coffee promoted bone health by improving articular cartilage constituents in rats exposed to both cadmium and lead [51]. Although we found no prominent association between coffee intake and bone health, more studies are needed to assess the effects of coffee because of the complexity of its constituents.

### 4.2. Effects of Tea on Bone Health

Studies have reported inconsistent findings regarding the association between tea intake and BMD. A study indicated that a higher BMD was associated with a higher level of tea intake [52]; this finding is not consistent with that of our study. The biological mechanisms underlying the effects of tea on bone health have been investigated in several studies. Polyphenols [53], may increase the level of circulating vitamin D in the body [54], which accelerates calcium excretion [50] and reduces osteoblast apoptosis [55], thereby enhancing cell proliferation and differentiation [56]. Flavonoids exert minor estrogenic effects that may promote bone health. Furthermore, epigallocatechin-3-gallate, a key flavonoid compound, induces the apoptosis of osteoclasts and enhances the alkaline phosphatase activity of osteoblast-like cells, both of which may increase bone mineralization [57].

We observed no effects of tea on the bone health of postmenopausal women. Similar findings were reported in an animal study [58], in which polyphenol supplementation was more beneficial (in terms of BMD) for estrogen-adequate middle-aged female rats than for estrogen-deficient (ovariectomized) middle-aged female rats; this finding is attributable to the binding affinities of epigallocatechin-3-gallate and epicatechin-3-gallate to estrogen receptors [59]. The effects of tea intake on hip fracture risk were noted to be nonsignificant in the present study; further robust studies are needed validate this finding.

Tea has many constituents in addition to caffeine. In the present meta-analysis, the mean BMD of individuals who consumed tea was higher than that of those who consumed coffee; however, this difference was nonsignificant. Although several studies have explored the effects of tea constituents on bone health, whether the complex constituents of tea negatively affect bone health must be investigated further.

### 4.3. Strengths

Our study has three major strengths. Firstly, earlier studies [6,27] did not specify the threshold for high-intake versus low-intake classification, thus the likelihood of misclassification between exposure and control groups for meta-analysis cannot be ignored. For example, 2 cups/day of coffee intake can be defined as the low-level control group in one study but as a high-level exposure group in another. We resolved this problem by specifying the thresholds used for classification. Second, to the best of our knowledge, this study is the first meta-analysis to include studies that reported DXA-based BMD data instead of osteoporosis diagnosis; thus, we could prevent possible errors associated with the use of multiple measurement methods. Finally, this study may be the first meta-analysis to concurrently evaluate the effects of tea and coffee to comprehensively compare their associations with BMD and hip fracture risk.

Our study has three minor strengths. First, hip fracture-related studies included in our meta-analysis were cohort, but not case-control or observational studies, because a cohort design can ensure a robust validity of correlation analyses. Second, the sample size was high. Finally, no publication bias was observed in our meta-analysis.

### 4.4. Limitations

Our study has some limitations. First, the included studies varied in terms of covariates. In several studies, the effects of crucial confounders (e.g., calcium intake, physical activity level, and smoking) were not adjusted. Furthermore, instruments used for data collection varied across studies (e.g., diet habit questionnaires, dietary recall history, and food frequency questionnaires). Consequently, a high level of heterogeneity is likely. Second, the generalizability of our findings beyond the populations studied may be limited because the included studies were conducted in only a few countries. Most studies on the association of coffee intake with bone health were conducted in China and the United States. Furthermore, almost all studies on the association of tea on bone health were conducted in China. Third, because of limited data availability, dose–response analyses could not be performed to generate further evidence. Whether the studied populations consumed only coffee, only tea, or both beverages could not be determined. Because the levels of coffee and tea intake were assessed through self-reported questionnaires, differentiating between the intake of coffee or tea and that of other caffeinated beverages was difficult. Because some coffee drinkers could also be habitual tea drinkers, the possibility of an underestimation of the effects of caffeinated beverages on bone health cannot be ignored.

## 5. Conclusions

The results of our meta-analysis indicate that the daily intake of caffeinated beverages, such as coffee and tea, is not associated with BMD or hip fracture risk, particularly in postmenopausal women. However, owing to the heterogenicity and limitations pertaining to dose–response analyses, rigorously designed large-scale cohort studies are warranted to elucidate the effects of caffeinated beverages on bone health. Our findings help reduce the inconsistency in the current literature regarding the aforementioned association. Furthermore, this study may serve as a reference for future studies aimed at identifying the risk factor for osteoporosis.

## Figures and Tables

**Figure 1 medicina-59-01177-f001:**
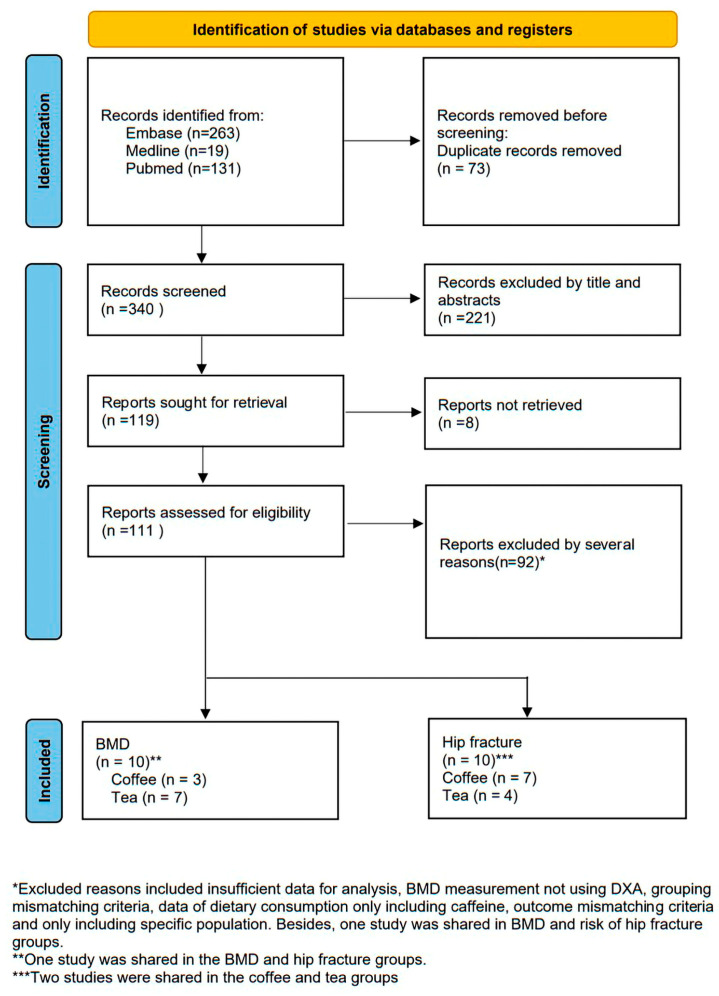
Flowchart of study selection.

**Figure 2 medicina-59-01177-f002:**
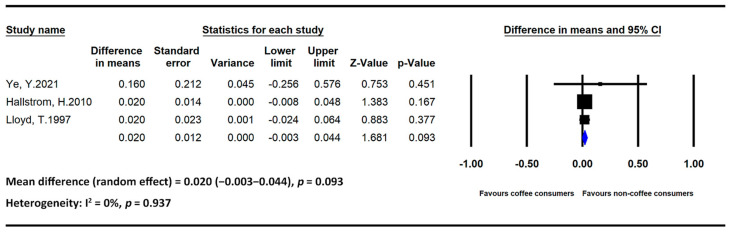
Statistics of studies regarding the effects of coffee intake on bone mineral density [34,35,36] (■: mean difference of each included studies; ◆: the pooled mean difference of included studies).

**Figure 3 medicina-59-01177-f003:**
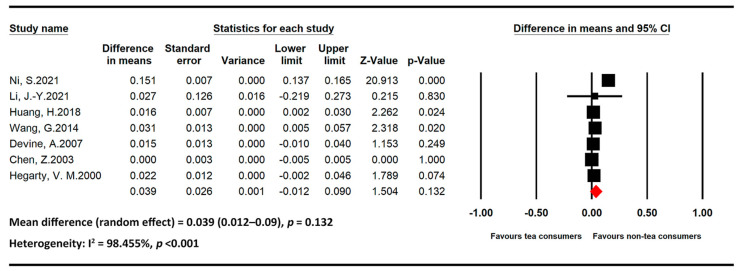
Statistics of studies regarding the effects of tea intake on bone mineral density [11,37,38,39,40,41,42] (■: mean difference of each included studies; ◆: the pooled mean difference of included studies).

**Figure 4 medicina-59-01177-f004:**
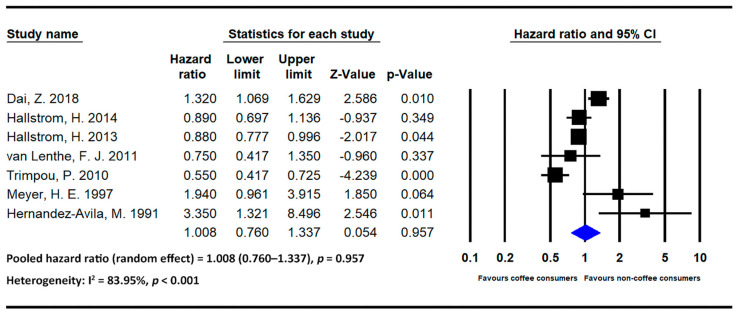
Statistics of studies regarding the effects of coffee on hip fracture risk [14,17,20,21,23,25,26] (■: hazard ratio of each included studies; ◆: the pooled hazard ratio of included studies).

**Figure 5 medicina-59-01177-f005:**
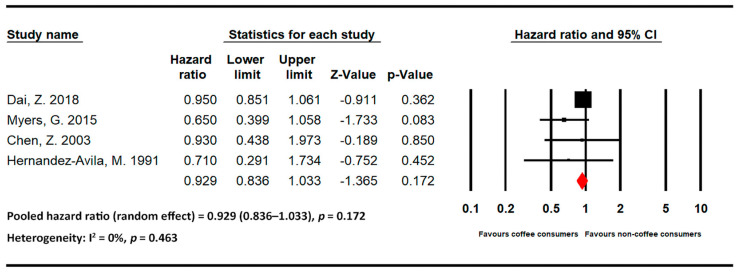
Statistics of studies regarding the effects of tea on hip fracture risk [11,24,25,26] (■: hazard ratio of each included studies; ◆: the pooled hazard ratio of included studies).

**Table 1 medicina-59-01177-t001:** Characteristics of studies included in our meta-analysis.

BMD Group							
Author, year	Country	Study design	Study population (age in years)	Groups	Number of participants		Bone mineral density mean difference
Coffee							
Ye, 2021 [34]	China	Cross-sectional	Postmenopausal women (47–90)	≥2 cups/day vs. 0 cups/day	648		0.16
Hallström, 2010 [35]	Sweden	Prospective cohort	Men and women (72)	>4 cups/day vs. 0–2 cups/day	717		0.02
Lloyd, 1997 [36]	United States	Cross-sectional	Postmenopausal White women (55–70)	≥5 cups/day vs. 0–2 cups/day	138		0.02
Tea							
Ni, 2021 [37]	China	Cross-sectional	Postmenopausal women (<80)	Drinkers vs. nondrinkers	1377		0.151
Li, 2021 [38]	China	Cross-sectional	Postmenopausal women	Daily drinkers vs. nondrinkers	947		0.027
Huang, 2018 [39]	China	Prospective cohort	Women (≥40)	≥7 times/week vs. ≤1 time/week	1495		0.016
Wang, 2014 [40]	China	Cross-sectional	Postmenopausal women (62.2519 ± 6.2837)	≥1 cup/day vs. 0 cup/day	680		0.031
Devine, 2007 [41]	Australia	Cross-sectional	Postmenopausal women (70–85)	≥5 cups/day vs. 0 cup/day	1027		0.015
Chen, 2003 [11]	United States	Prospective cohort	Postmenopausal women (50–79)	≥5 cups/day vs. <1 cup/day	91,465		0.000
Hegarty, 2000 [42]	United Kingdom	Cross-sectional	Postmenopausal women (65–76)	>6 cups/day vs. 0 cup/day	1256		0.048
**Hip Fracture Group**							
Author, year	Country	Design	Study population (age in years)	Groups	Number of participants	Number of hip fracture cases	Hazard ratio (95% confidence interval)
Coffee							
Dai, 2018 [25]	Singapore	Prospective cohort	Men and women (45–74)	≥4 cups/day vs. <1 cup/week	63,154	2502	1.32 (1.07–1.63)
Hallström, 2014 [23]	Sweden	Prospective cohort	Men (45–79)	≥4 cups/day vs. <1 cup/day	42,978	1186	0.89 (0.70–1.14)
Hallström, 2013 [17]	Sweden	Prospective cohort	Women (not available)	≥4 cups/day vs. <1 cup/day	61,433	3871	0.88 (0.78–1.00)
van-Lenthe, 2011 [21]	The Netherlands	Prospective cohort	Women (25–74)	≥3 cups/day vs. 0 cup/day	16,578	192	0.75 (0.42–1.36)
Trimpou, 2010 [19]	Sweden	Prospective cohort	Men (46–56)	≥5 cups/day vs. 0 cup/day	7495	451	0.55 (0.42–0.73)
Meyer, 1997 [14]	Norway	Prospective cohort	Men and women (35–49)	≥9 cups/day vs. <1 cup/day	39,787	213	Women: 1.94 (0.96–3.91) Men: 1.04 (0.37–2.94)
Hernandez-Avila, 1991 [28]	United States	Prospective cohort	Women (34–59)	≥4 cups/day vs. individuals who rarely drank coffee	84,484	65	3.35 (1.32–8.49)
Tea							
Dai, 2018 [25]	Singapore	Prospective cohort	Men and women (45–74)	Daily vs. less frequent than weekly	63,154	2502	0.95 (0.85–1.06)
Myers, 2015 [24]	Australia	Prospective cohort	Postmenopausal women (>75)	≥3 cups/day vs. ≤1 cup/day	1188	129	0.65 (0.40–1.06)
Chen, 2003 [11]	United States	Prospective cohort	Postmenopausal women (50–79)	≥5 cups/day vs. <1 cup/day	91,465	386	0.93 (0.44–1.98)
Hernandez-Avila, 1991 [26]	United States	Prospective cohort	Women (34–59)	≥2 cups/day vs. individuals who rarely drank tea	84,484	65	0.71 (0.29–1.73)

## Data Availability

The data presented in this study are openly available in PubMed, reference number [14,17,20,21,23,24,25,26,35,36,37,38,39,40,41,42].

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
