# Peer review of "Association of Coffee and Tea Intake with Bone Mineral Density and Hip Fracture: A Meta-Analysis"

_medicina, 2023, doi:10.3390/medicina59061177_

Round 1

Reviewer 1 Report

The study is focused on the meta-analysis of coffee and tea intake on bone mineral density and hip fracture. The authors analyzed 413 articles, 20 of which met the selected criteria and were selected for meta-analysis. The paper contributes to filling the void in the current state of knowledge. The work is written in a clear and coherent way. The results were analyzed according to appropriate methods. The discussion is carried out correctly. The conclusions presented are consistent. The tables are well described. References typical and actual. Most of the shortcomings are described in the study limitation section.

Shortcomings:

-          Please add the explanation of the abbreviation MeSH line 69

-          References to figures in the text are incorrect in lines 171 and 192

-          The references section should be edited in accordance with the requirements of the publisher

Reviewer 2 Report

The authors of this study were aimed, based on meta-analysis of prospective cohort and cross-sectional studies, we investigated the association of coffee and tea intake with bone mineral density (BMD) and hip fracture risk.

The issue is important, moreover they conduct comprehensive results and discussion. Therefore, I have no further comments against the manuscript.

Reviewer 3 Report

This meta-analysis offers a useful contribution to the field of bone density research -which has not always been deeply investigated concerning lifestyle in the last few years. The methodological approach for systematic reviews is correct, according to the PRISMA checklist. Also, the inclusion criteria were carefully selected. The choice to only include studies concerning DEXA data is methodologically appreciable -although it could have been interesting to verify these associations from studies adopting also other devices, such as ultrasonometers. Evidences are well documented and discussed, and provide important contributions in the field. Also, the limitations enounced by the authors are correct and offer interesting hints for future research. 
